# Ursolic Acid Alleviates Neuroinflammation after Intracerebral Hemorrhage by Mediating Microglial Pyroptosis via the NF-κB/NLRP3/GSDMD Pathway

**DOI:** 10.3390/ijms241914771

**Published:** 2023-09-30

**Authors:** Pan Lei, Zhiyang Li, Qiuwei Hua, Ping Song, Lun Gao, Long Zhou, Qiang Cai

**Affiliations:** Department of Neurosurgery, Renmin Hospital of Wuhan University, Wuhan 430060, China; 2021283020190@whu.edu.cn (P.L.); 2018283020174@whu.edu.cn (Z.L.); 2022203020026@whu.edu.cn (Q.H.); songping1201@163.com (P.S.); lungao@whu.edu.cn (L.G.)

**Keywords:** ursolic acid, neuroinflammation, cerebral hemorrhage, microglial, pyroptosis

## Abstract

The neuroinflammatory response after intracerebral hemorrhage (ICH) causes a large amount of neuronal loss, and inhibiting the inflammatory response can improve the prognosis. In previous laboratory studies and clinical trials, ursolic acid (UA) inhibited the inflammatory response, but whether it can be administered to inhibit the neuroinflammatory response after cerebral hemorrhage is unknown. The aim of this study was to investigate the effects of ursolic acid after cerebral hemorrhage. Online databases were used to obtain potential therapeutic targets of ursolic acid for the treatment of cerebral hemorrhage, and possible mechanisms were analyzed by KEGG, GO, and molecular docking. A rat model of cerebral hemorrhage was established using collagenase, and an in vitro cerebral hemorrhage model was constructed by adding hemin to BV2 cell culture medium. Enzyme-linked immunosorbent assay (ELISA), Western blotting (WB), immunofluorescence, TUNEL staining, and calcein/PI staining were used to investigate the degree of microglial M1 polarization, changes in the levels of inflammatory factors, activation of the NF-κB pathway, and changes in the indicators of cellular death after ursolic acid treatment. In addition, phorbol 12-myristate 13-acetate (PMA) was used to activate the NF-κB pathway to verify that ursolic acid exerts its anti-neuroinflammatory effects by regulating the NF-κB/NLRP3/GSDMD pathway. Network pharmacology and bioinformatics analyses revealed that ursolic acid may exert its therapeutic effects on cerebral hemorrhage through multiple pathways. Together, in vivo and in vitro experiments showed that ursolic acid inhibited microglial M1 polarization and significantly reduced the levels of p-NF-κB, GSDMD-N, cleaved caspase-1, TNF-α, IL-6, and IL-1β, which were significantly inhibited by the use of PMA. Ursolic acid inhibits microglial pyroptosis via the NF-κB/NLRP3/GSDMD pathway to alleviate neuroinflammatory responses after cerebral hemorrhage.

## 1. Introduction

Cerebral hemorrhage is a disease with high mortality and disability rates, and while the existing treatments greatly reduce the mortality rate, their effects on improving prognosis are very limited [1,2,3]. This is mainly due to the loss of neurons after cerebral hemorrhage, which results in a loss of neurological function that cannot be restored [4]. An increasing number of studies have revealed that secondary injury after cerebral hemorrhage can lead to a large amount of neuronal loss, and neuroinflammation plays a key role in this process [5,6]. Thus, inhibiting neuroinflammation has become an attractive possibility as an effective means to improve the prognosis of cerebral hemorrhage [7].

The activation of many immune cells after cerebral hemorrhage leads to an intense inflammatory response, and as more reports have been released, it is gradually being realized that microglia-mediated neuroinflammation is the most important factor in inflammatory injury after cerebral hemorrhage [8,9]. Microglia are macrophages that reside in the brain and can be transformed from a resting state to an M1 or an M2 phenotype depending on stimulation [10]. M1-type glial cells can play a defensive role by phagocytosing and clearing cellular debris, but when overactivated, M1-type glial cells can rapidly produce many inflammatory factors, which can lead to sustained brain injury [11,12]. M2-type glial cells can phagocytose hematoma degradation products to reduce the toxicity of the hematoma and can secrete anti-inflammatory mediators to inhibit the inflammatory response [13]. Reasonable regulation of the ratio of M1 to M2 glial cells is important for the development of the neuroinflammatory response after cerebral hemorrhage and is also an important target for the inhibition of neuroinflammation [14].

Ursolic acid is a pentacyclic triterpene found in a variety of plants and is widely used in the treatment of many diseases [15,16]. Ursolic acid has been found to inhibit inflammatory responses in previous clinical trials and laboratory studies [17,18]. Its anti-inflammatory effects on traumatic brain injury and subarachnoid hemorrhage are also significant [19]. Several experiments have demonstrated that ursolic acid can effectively inhibit the inflammatory response after subarachnoid hemorrhage and can improve prognosis, but whether ursolic acid can inhibit neuroinflammation and improve prognosis in parenchymal hemorrhage is not yet known [20,21].

In this study, we predicted the therapeutic targets of ursolic acid using several online databases, made a conjecture that ursolic acid may inhibit neuroinflammation after cerebral hemorrhage through multiple pathways based on bioinformatics methods, and finally validated our conjecture in detail with in vivo and in vitro experiments. All the experimental steps are shown in Figure 1.

## 2. Results

### 2.1. Network Analysis Suggests That Ursolic Acid May Treat Cerebral Hemorrhage via Multiple Targets

Ursolic acid (Figure 2A) target genes were acquired through the Superpred, Swisstarget, Sae, Targetnet, and Pharmmapper websites, and we obtained a total of 423 valid genes after deleting duplicate values. Subsequently, we searched for brain hemorrhage-related genes through the Genecards, NCBI, and OMIM databases, and we obtained a total of 1338 genes after deleting duplicate values. After taking overlapping ursolic acid target genes and brain hemorrhage marker genes (Figure 2B), a total of 124 valid genes were obtained (Appendix A).

To explore the interactions of these 124 genes, they were imported into the String database, and a minimum required interaction score of 0.9 was used to obtain a PPI network, which was subsequently analyzed in depth using Cytoscape software (Figure 2C).

To further clarify the potential therapeutic effect of ursolic acid on cerebral hemorrhage, the genes were imported into the DAVID database. *P* < 0.05 was used to indicate statistical significance, and 10 entries from GO enrichment results and KEGG enrichment results were selected for presentation. According to the GO enrichment results, the UA-ICH target may be a cellular component of “NF-κB complex”, “lysosome”, “transcription factor complex”, and “regulation of microglial cell activation.” It may also be involved in biological processes such as “regulation of microglial cell activation”, “regulation of cytokine production involved in inflammatory response”, and “I-κB kinase/NF-κB signaling”. The KEGG enrichment results suggest that the mechanism of UA treatment for cerebral hemorrhage may involve “HIF-1 signaling pathway”, “PI3K/Akt signaling pathway”, “NF-κB signaling pathway” and other pathways. The series of results suggests that ursolic acid may intervene in cerebral hemorrhage through multiple pathways (Figure 2D–F).

### 2.2. Ursolic Acid Improves Neurological Deficits in Rats with Experimental Cerebral Hemorrhage

A cerebral hemorrhage model using collagenase injection into the right basal ganglia region of rats was established to investigate the in vivo efficacy of ursolic acid. To determine the appropriate dose of ursolic acid, we set doses of 10, 20, and 40 mg/kg and evaluated the neurological deficits of the rats by using the corner test and the mNss score 1, 3, and 7 days after cerebral hemorrhage [22]. The results showed that the 20 mg/kg dose significantly improved neurological deficits, and more significant therapeutic effects were not seen as the dose was increased (Figure 3C,D).

Based on this finding, we chose a moderate dose for the study. To further investigate the neuroprotective effect of ursolic acid on rats with cerebral hemorrhage, we selected brain tissue sections from 3 days after hemorrhage and performed TUNEL staining (Appendix A) and Nissl staining. HE staining showed edema and necrosis of the cells in the tissue around the hematoma after cerebral hemorrhage (Figure 3B). Nissl staining and TUNEL staining demonstrated that ursolic acid could reduce the loss of neurons around the hematoma after cerebral hemorrhage at an effective therapeutic dose (Figure 3F–H). The WB results suggested that UA treatment significantly reduced the expression of cleaved caspase-3 and Bax and upregulated the expression of Bcl-2 in the perihematomal tissues compared with the vehicle group and the ICH group (Figure 3I–K). Meanwhile, to exclude the possible effect of ursolic acid on collagenase, we observed the hematoma morphology one day after the model was established and found that feeding ursolic acid in advance did not lead to significant changes in the hematoma volume, which also showed that the neuroprotective effect of ursolic acid was not achieved by reducing the hematoma volume (Figure 3A,E).

### 2.3. Ursolic Acid Protects the Blood–Brain Barrier

To further investigate the possible neuroprotective effects of ursolic acid, we examined the integrity of the blood–brain barrier after ursolic acid treatment. First, wet and dry weight measurements were performed on rat brain tissue three days after cerebral hemorrhage, and it was found that the water content of the brain tissue ipsilateral to the hematoma after cerebral hemorrhage was significantly higher than that on the contralateral (healthy) side and that after ursolic acid treatment, the water content of the brain tissue around the hematoma decreased significantly (Figure 4D).

To make the results more accurate, the Evans blue (EB) permeation assay was also used to assess the blood–brain barrier integrity (Figure 4C). We also examined the expression of occludin and claudin-5, two classic intercellular tight junction proteins, and UA inhibited the decrease in the expression of these two proteins compared with the vehicle and ICH groups (Figure 4E–G). ZO-1 immunofluorescence likewise showed that UA inhibited blood–brain barrier disruption after cerebral hemorrhage (Figure 4A,B).

### 2.4. Ursolic Acid Inhibits the Inflammatory Response and Decreases Microglial M1 Polarization

In view of the target prediction results suggesting that ursolic acid may have a role in inhibiting the neuroinflammatory response after cerebral hemorrhage and because the anti-inflammatory effect of ursolic acid has been confirmed by many experiments, we explored whether ursolic acid could inhibit the inflammatory response around the hematoma after cerebral hemorrhage. The WB results showed that after treatment with UA, the levels of IL-6 and TNF-α in the brain tissues around the hematoma were significantly decreased (Figure 5A–G).

Microglia, which are resident macrophages in the brain, differentiate into M1 and M2 phenotypes upon stimulation, of which the M1 type has a strong proinflammatory effect. We labeled M1-type microglia using Iba1 and iNOS and found that the number of M1-type microglia was significantly increased in the vehicle group and the ICH group compared with the sham group, and UA effectively reversed this trend (Figure 5H).

### 2.5. Ursolic Acid Treatment Reduces Cerebral Hemorrhage-Induced Microglial Cell Pyroptosis In Vivo

Microglial pyroptosis causes the release of intracellular inflammatory factors, which can exacerbate the inflammatory response after cerebral hemorrhage. Inhibition of microglial pyroptosis after cerebral hemorrhage has been reported to reduce neuroinflammatory responses and improve prognosis [23]. To confirm whether ursolic acid can inhibit cerebral hemorrhage-induced cellular pyroptosis after in vivo administration, we examined the expression of proteins related to cellular pyroptosis in brain tissues around the hematoma, including GSDMD and GSDMD-activated proteins (GSDMN-N), caspase-1 and cleaved caspase-1, IL-1β and the activated form of IL-1β. Western blotting results showed that the expression levels of cleaved-caspase-1\GSDMD-N\IL-1β/ASC were significantly higher in the ICH group and vehicle group than in the sham groups, a result that suggests that cerebral hemorrhage can significantly induce cellular pyroptosis in tissues surrounding hematomas and that ursolic acid can block this process (Figure 6A–F).

NLRP3 has been shown to be closely related to cell pyroptosis in previous studies [24], and to further confirm that ursolic acid can inhibit microglial focalization after cerebral hemorrhage, we also used immunofluorescence technology to detect the fluorescence intensity of NLRP3 in Iba1-positive cells. Ursolic acid treatment significantly reduced NLRP3 fluorescence intensity in Iba1-positive cells compared with that in the vehicle group and the ICH group (Figure 6H,G). These results confirm that ursolic acid treatment can significantly inhibit cerebral hemorrhage-induced microglial pyroptosis.

### 2.6. Ursolic Acid Inhibits the Microglial Cell Inflammatory Response and Pyroptosis Stimulated by Hemin In Vitro

To further investigate the mechanism by which ursolic acid inhibits microglial pyroptosis after cerebral hemorrhage, we first generated a cerebral hemorrhage cell model by adding hemin to the BV2 cell culture medium. The CCK-8 assay results showed that at 60 µM, the viability of BV2 cells decreased by approximately 50%, so we chose 60 µM as the test concentration (Figure 7A). We also explored the possible harmful effects of ursolic acid on BV2 cells using the CCK-8 assay and chose 2 µM as an appropriate dose of ursolic acid to be used (Figure 7B). Subsequently, we set up sham, hemin, vehicle, and UA groups (Figure 7C).

First, to investigate whether UA could inhibit the M1 polarization of BV2 cells caused by hemin stimulation, we detected the M1-type glial cell marker proteins CD32 and CD86 by Western blotting experiments. The results confirmed that hemin treatment could significantly increase the expression of CD32 and CD86 in BV2 cells, and the levels of CD32 and CD86 in the UA group were significantly lower than those in the vehicle group and the hemin group. Subsequently, we detected the level of each inflammatory factor under hemin stimulation by ELISA. The results showed that hemin caused an increase in IL-1β/TNF-α/IL-6 levels secreted by BV2 cells, and UA reduced the secreted levels of inflammatory factors compared to the vehicle and hemin groups (Figure 7D–F). Meanwhile, the results of Western blotting experiments showed that hemin treatment of BV2 cells led to an increase in the expression of GSDMD-N/cleaved caspase-1/IL-1β, and UA treatment reversed this trend (Figure 7G–M). These results suggest that UA can inhibit hemin-induced M1 polarization in BV2 cells and similarly attenuate the hemin-induced inflammatory response and cellular pyroptosis.

### 2.7. UA Attenuates Microglial Focal Death after Cerebral Hemorrhage by Inhibiting the NF-κB Pathway

Activation of the NF-κB signaling pathway can lead to NLRP3 activation [25], and KEGG and GO enrichment analyses suggested that this pathway may be one of the therapeutic targets of ursolic acid. To verify this possibility, we first performed molecular docking, and the results showed that ursolic acid can freely bind to IκB (Figure 8A) and P65 (Figure 8B) (free energy < 0 kcal for each, Appendix A).

To further verify whether ursolic acid could inhibit the activation of the NF-κB pathway in the in vivo experiments, expression of key NF-κB pathway markers was measured. In the in vivo experiments, the expression of phosphorylated IκBα was significantly increased in the vehicle and ICH groups compared with the sham group, and the content of p-P65 was significantly lower in the UA group than in the vehicle and ICH groups (Figure 8F–H). In the in vitro experiments, we extracted intranuclear and cytoplasmic proteins by using a nuclear and cytoplasmic protein extraction kit and measured the above indexes. The results showed that after hemin stimulation, the expression of phosphorylated IκBα in BV2 cells increased significantly, and NF-κB P65 underwent nuclear translocation. Compared with the hemin and vehicle groups, the intranuclear NF-κB content in the UA group decreased, intracytoplasmic IκBα content increased, and p-IκBα content decreased (Figure 8C–E).

### 2.8. PMA Abrogates the Role of UA in Attenuating Microglial Pyroptosis

To further clarify whether ursolic acid inhibits microglial pyroptosis after cerebral hemorrhage by inhibiting the activation of the NF-κB pathway, we used PMA, an NF-κB pathway agonist, to conduct an in vitro experimental study.

Compared with the hemin + vehicle group, the expression of p-IκBα, p-p65, NLRP3, IL1β, ASC, cleaved caspase-1, and GSDMD-N was significantly increased in the Hemin + PMA group, and the expression of these proteins was significantly decreased after the use of UA. The results of PI staining (Appendix A) showed that the administration of PMA significantly increased the proportion of pyroptotic cells, and UA significantly reversed this change. The results suggest that ursolic acid suppresses the NF-κB/NLRP3 pathway to reduce microglial pyroptosis after cerebral hemorrhage (Figure 9A–H).

Similarly, we conducted in vivo experiments. First, we measured TNFα, IL-6, and IL-1β levels around the hematoma by ELISA (Figure 9I–K), and the results showed that concurrent use of PMA with UA reduced the inflammatory factor levels around the rat brain tissue compared to PMA alone. Immunofluorescence likewise showed that the changes in GSDMD content induced by PMA could be counteracted by the use of UA (Figure 9L–M). These results suggest that UA alleviates the inflammatory response after cerebral hemorrhage in part by inhibiting NF-KB pathway activation.

### 2.9. In Vivo Biosafety Assessment of Ursolic Acid

To investigate whether ursolic acid ingestion causes damage to rats, an in vivo toxicity test was conducted.

The results showed that continuous feeding of rats with a dose of 20 mg/kg did not result in death. Compared with the sham group, the vehicle and UA groups showed significant weight gain, and there was no significant difference in body weight between the UA and vehicle groups (Figure 10A).

Meanwhile, Hb (Figure 10B), WBC (Figure 10C), RBC (Figure 10D), ALB (Figure 10E), FBS (Figure 10F), BUN (Figure 10I), Cr (Figure 10J), ALT (Figure 10K), and AST (Figure 10L) indexes were all within the normal range, indicating that continuous feeding of ursolic acid at therapeutic doses did not damage the liver and kidney functions of rats. However, it is worth noting that the large amount of fat intake due to the use of rapeseed oil as a solvent resulted in elevated TG (Figure 10G) and CHOL (Figure 10H) in the vehicle and UA groups compared to the sham group, but there was no significant difference between these two groups.

## 3. Discussion

Cerebral hemorrhage is a class of diseases with high disability and mortality rates, and its long-term prognosis is poor [26]. To date, surgical or medical treatments are focused on preventing rebleeding and reducing intracranial pressure in time to reduce the risk of cerebral herniation. However, due to the nonregenerative nature of nerve cells, improving neurological deficits after cerebral hemorrhage is still a major challenge [27,28]. Therefore, minimizing neuronal loss has become an important goal for improving prognosis after cerebral hemorrhage. Neuronal damage after cerebral hemorrhage mainly comes from two aspects: the primary damage caused by hematoma and the secondary damage caused by the toxic effects of hematoma degradation products. Primary damage is unavoidable, so current treatments for cerebral hemorrhage mainly focus on reducing secondary damage after cerebral hemorrhage [29].

Neuroinflammation after cerebral hemorrhage is the main cause of neuronal death secondary to injury, and how to effectively inhibit the neuroinflammatory response is an important research direction for the treatment of cerebral hemorrhage [30]. Microglia, as resident intracranial macrophages, are the main initiators of the inflammatory response [31]. After cerebral hemorrhage, microglia rapidly transform into the M1 phenotype and release inflammatory factors such as TNF-α, IL-1β, and IL-6 [32,33]. These inflammatory factors further activate the NF-κB pathway in microglia, which leads to the activation of NLRP3 and the cleavage of caspase-1, which can subsequently cleave GSDMD to form active N-terminal GSDMD and C-terminal GSDMD. N-terminal GSDMD can adhere to cells and cause cell perforation [34]. The perforation of microglia leads to an accelerated release of intracellular inflammatory factors, and the leakage of inflammatory factors repeats the above process, leading to an increased neuroinflammatory response [23,35].

Ursolic acid is widely used as an anti-inflammatory drug, and many biological experiments have demonstrated that ursolic acid can play an anti-inflammatory role in a variety of diseases, including both oncological and non-oncological diseases [36,37]. Clinical trials are now being carried out to verify its efficacy and safety [38]. However, to date, whether ursolic acid can inhibit neuroinflammation after parenchymal hemorrhage has not been reported.

In our experiments, to investigate the potential anti-inflammatory effects of ursolic acid after a parenchymal brain hemorrhage, a rat model of collagenase cerebral hemorrhage and an in vitro cerebral hemorrhage model of hemin-stimulated BV2 cells, which are currently widely used as disease models, were constructed. Possible targets of ursolic acid for the treatment of cerebral hemorrhage were predicted using several online sites. Through in vivo experiments, we found that the neurological deficits of rats with cerebral hemorrhage were significantly improved after the administration of ursolic acid, and the water content of the brain tissue around the hematoma and the permeability of the blood–brain barrier were significantly decreased, which all showed that ursolic acid was beneficial to rats with cerebral hemorrhage. To further investigate the mechanism of ursolic acid in the treatment of cerebral hemorrhage, we compared the levels of inflammatory factors in the brain tissues of rats from different groups and found that ursolic acid could effectively inhibit the inflammatory reaction after cerebral hemorrhage and inhibit the conversion of microglial cells to the M1 phenotype. Concurrently, we found that ursolic acid could effectively inhibit microglial pyroptosis by immunofluorescence. Subsequently, to understand the effect of ursolic acid on microglia in more detail, we simulated cerebral hemorrhage in vitro by adding hemin to BV2 cell culture medium. The results suggest that ursolic acid can effectively inhibit P65 nuclear translocation, which in turn inhibits NLRP3 activation and prevents the onset of cellular pyroptosis. However, the protective effect of ursolic acid disappeared when we used an NF-κB pathway activator, which suggests that the inhibitory effect of ursolic acid on neuroinflammation may partly depend on the inhibition of NF-κB pathway activation.

Our experiments demonstrated for the first time that ursolic acid inhibits neuroinflammation after cerebral hemorrhage, partially through the NF-κB/NLRP3 pathway. The main limitation of this study is that the in vitro cell model cannot fully simulate in vivo cerebral hemorrhage. There are currently two main types of cerebral hemorrhage models, one with collagenase injection to induce hemorrhage and one with direct blood injection, although both models were widely used before this, not only to study neuroinflammation. Refs. [4,39] However, in our present study, we used collagenase injection into the striatum to establish brain hemorrhage in order to ensure the stability of hematoma volume in rats and also demonstrated that ursolic acid does not interfere with collagenase-induced hemorrhage. However, it is still difficult to determine whether ursolic acid can similarly antagonize the neuroinflammatory response in a model of cerebral hemorrhage using injected blood. This is what we need to accomplish next.

At the same time, although we were concerned about the possible in vivo biotoxic response of ursolic acid and tested mainly for blood biochemical indices and liver and kidney function, our tests appeared to be very insufficient in comparison with the complex toxicological tests. In the future, additional indicators need to be included to assess the biosafety of ursolic acid more fully.

This experiment confirms that ursolic acid inhibits neuroinflammation after cerebral hemorrhage partly through inhibition of the NF-κB pathway and that neuronal loss can be attenuated through inhibition of neuroinflammation, but it remains to be further demonstrated whether UA can directly produce therapeutic effects on neurons.

## 4. Materials and Methods

### 4.1. Analysis of the Overlap of Ursolic Acid Therapeutic Targets and Cerebral Hemorrhage Markers

#### 4.1.1. Acquisition of Therapeutic Targets for Cerebral Hemorrhage

We searched the OMIM database (https://www.omim.org, 25 December 2022), NCBI database (www.ncbi.nlm.nih.gov, 25 December 2022), and Genecards database (https://www.genecards.org, 25 December 2022) for cerebral hemorrhage-related genes using the search terms “cerebral hemorrhage” and “intracerebral hemorrhage”. Duplicates were removed and combined as therapeutic target genes for cerebral hemorrhage.

#### 4.1.2. Ursolic Acid Therapeutic Target Prediction

Ursolic acid was searched as a search term sequentially in Superpred database(https://prediction.charite.de, 28 December 2022), Swisstarget database (http://www.swisstargetprediction.ch, 28 December 2022), SEA database (http://edbc.org, 28 December 2022), targetnet (http://targetnet.scbdd.com, 28 December 2022) [40] and Pharmmapper database (http://lilab-ecust.cn, 28 December 2022) [41] to obtain the therapeutic targets of ursolic acid. The therapeutic targets of ursolic acid were obtained after the process of removing duplicate values.

#### 4.1.3. Ursolic Acid Therapeutic Targets and Cerebral Hemorrhage Marker Overlap

The obtained cerebral hemorrhage-related markers and ursolic acid-related targets were imported into the Venny online website to obtain potential targets of ursolic acid for the treatment of cerebral hemorrhage.

#### 4.1.4. Protein–Protein Interaction Network Construction

The intersecting target genes were imported into the STRING database (https://cn.string-db.org, 3 January 2023), and the minimum required interaction score was set to 0.9. The protein interaction networks obtained were imported into Cytoscape software (https://cytoscape.org, 3 January2023), and then the Hub genes were screened and sorted according to the CytoHubba plug-in.

#### 4.1.5. Enrichment Analysis of Genes Overlapping Cerebral Hemorrhage and Ursolic Acid

GO and KEGG analyses of overlapping genes were performed using the DAVID (https://david.ncifcrf.gov, 3 January 2023) website to obtain the GO and KEGG signaling pathways of genes related to the ursolic acid treatment of cerebral hemorrhage.

### 4.2. Reagents

Ursolic acid (SU8020) and hemin solution (0.5 mg/mL, H8132) were provided by Solarbio Life Sciences (Beijing, China). Rapeseed oil (C116025) and Evans blue dye (E104208) were purchased from Aladdin (Shanghai, China). Collagenase IV (#2091) was purchased from BioFroxx (Guangzhou, China). Phorbol 12-myristate 13-acetate (PMA, HY-18739) was purchased from MedChemExpress (MCE, Monmouth Junction, NJ, USA).

Anti-Bax (SC-7480), anti-GSDMD (SC-393581), and anti-caspase-1 antibodies (SC-392736) were purchased from Santa Cruz Biotechnology (Santa Cruz, CA, USA). Anti-cleaved caspase-1 antibody (#89332S), anti-cleaved caspase-3 antibody (#9664S), anti-cleaved GSDMD antibody (#36425S), and anti-p-P65 antibody (#3033S) were purchased from Cell Signaling Technology (Danvers, MA, USA). Anti-GAPDH antibody (60004-1-lg), anti-claudin-5 antibody (29767-1-AP), anti-occludin antibody (66378-1-lg), anti-CD32 antibody (15625-1-AP), anti-CD86 antibody (13395-1-AP), anti-ASC antibody (10500-1-AP), anti-NLRP3 antibody (68102-1-lg), anti-Histone H3 antibody (68345-1-lg), anti-P65 antibody (10745-1-AP), anti-ZO-1 antibody (21773-1-AP), anti-iNOS antibody (80157-l-RR), anti-β actin antibody (81115-1-RR), and anti-iba1 antibody (10904-l-AP) were purchased from Proteintech (Wuhan, China). Anti-COX2 antibody (A1253), anti-IL-6 antibody (A0286), anti-TNF-α antibody (A11534), and anti-IL-1β antibody (A20529) were purchased from ABclonal Technology (Wuhan, China). Anti-Bcl-2 antibody (WL01556), anti-IκBα antibody (WL01936), and anti-p-IκBα antibody (WL02495) were purchased from Wanleibio (Shenyang, China).

Horseradish peroxidase (HRP)-conjugated secondary antibodies and immunofluorescence secondary antibodies were purchased from Proteintech (Wuhan, China).

Elisa kits for IL-6 (P1326, P1328), TNF-α (PT512, PT516), and IL-1β (P1301, P1307), Calcein/PI Cell Viability Assay Kit (C2015S), and Cell Counting Kit-8 (C0038) were bought from Beyotime (Shanghai, China).

### 4.3. Animal Experiments

#### 4.3.1. Animals

A total of 194 SD rats (Hubei Bainter Biotechnology Co., Ltd., Wuhan, China) were used in this experiment, of which 20 died due to anesthesia or postoperative respiratory obstruction and hemorrhagic displacement. Rats were housed in the Animal Experiment Center of Wuhan University People’s Hospital, where they were maintained at appropriate temperature and humidity and supplied with adequate food and water. This study was approved by the Laboratory Animal Ethics Committee of Wuhan University People’s Hospital (ID number: WDRM20220301C).

#### 4.3.2. Modeling of Cerebral Hemorrhage

As previously reported [4], the rats were anesthetized using 1% pentobarbital sodium solution at 40 mg/kg and were fixed in the stereotaxic apparatus after removing the head hair. After disinfection, the skin was incised, the periosteum was exposed, and a hole was drilled 0.1 mm behind the sagittal suture and 3.5 mm to the right of the midline. Then, 2 µL of collagenase IV (12.5 U/mL) was extracted from the vial, and the needle was injected slowly to a distance of 6 mm. The needle was left in place for 10 min, bone wax was used to close the hole, and the wound was sutured and disinfected again.

In order to improve the bioavailability of ursolic acid in vivo, drawing on previous studies, we chose to use rapeseed oil as a solvent for ursolic acid [42]. All the groups except the vehicle group were given ursolic acid intervention by gavage 3 days in advance, and the vehicle group was fed only rapeseed oil. The sham group only punctured the brain tissue with a microsyringe without collagenase injection, while the ICH group, the UA group, and the vehicle group received an injection with collagenase.

#### 4.3.3. Neurologic function assessment

As previously described, neurological function was assessed in each group of rats on days 1, 3, and 7 after cerebral hemorrhage, and mNss scores and corner-turning tests were carried out [43,44].

#### 4.3.4. Paraffin Section Production

After the rats were deeply anesthetized using sodium pentobarbital, the chest cavity was cut open to expose the heart, and the inferior vena cava was clipped while 0.9% sodium chloride solution containing heparin was injected into the left ventricle until clarified fluid flowed out of the inferior vena cava or the liver turned yellow. The skull was carefully peeled off, and intact brain tissue was removed and placed in 4% paraformaldehyde fixative for 48 h. Subsequently, the brain was put into a dehydrator for gradient dehydration and dipped in wax, and finally, the wax block was put into a slicer to cut 5 μm slices for subsequent experiments.

#### 4.3.5. Measurement of Brain Water Content

The complete brain tissue was taken from rats after deep anesthesia and divided into three parts: hemorrhage side cerebral hemisphere, healthy side cerebral hemisphere, and cerebellum. The three brain tissues were placed on an electronic balance and weighed; the value was recorded as the wet weight of the brain tissues; and the three brain tissues were placed in an oven to be dried for 48 h at 60 °C. Brain water content = (wet weight − dry weight)/wet weight × 100%.

#### 4.3.6. TUNEL Staining

After gradient deparaffinization of paraffin sections, the intensity of TUNEL fluorescence staining in brain tissues at each time point was determined using a TUNEL assay kit according to the manufacturer’s instructions, and quantitative analysis was performed using ImageJ software (V 1.8.0).

#### 4.3.7. Nissl Staining

The paraffin sections were deparaffinized and processed as previously described, stained with Nissl stain, rinsed clean, and then viewed under the microscope. Nissl-positive microsomes were counted.

#### 4.3.8. Immunofluorescence Staining

Brain tissue sections were deparaffinized and incubated with primary and secondary antibodies according to the experimental protocol, and fluorescence was observed under a fluorescence microscope. The fluorescence intensity was measured with ImageJ software (V 1.8.0).

#### 4.3.9. In Vivo Biosafety Assessment of Ursolic Acid

Three groups of four rats were set up in total. The sham group was fed daily with a gavage needle inserted into the esophagus without any medication, the vehicle group was fed rapeseed oil nasally at 10 mL/kg, and the UA group was fed ursolic acid dissolved in rapeseed oil at the therapeutic dose nasally at 10 mL/kg. These manipulations were carried out once a day. The body weight of each group was recorded, and blood was collected on the 14th day for blood biochemistry tests.

### 4.4. Cellular Experiments

#### 4.4.1. BV2 Cell Culture and Processing

BV2 microglial cells were purchased from Procell (Wuhan, China), cultured in a BV2-specific medium (Procell, Wuhan, China) at 37 °C and 5% CO_2_, and treated with drugs added at appropriate times according to the drug instructions.

#### 4.4.2. Cell Counting kit-8 (CCK-8) Assay

As previously described, different concentrations of UA or hemin were added to the BV2 cell culture medium and incubated for 24 h. Reagents were added according to the manufacturer’s instructions, and absorbance was measured at 450 nm. Cell survival rate (%) = (A the experimental group/A the control group) × 100%. The experiment was independently repeated 3 times [45].

#### 4.4.3. Calcein/PI Staining

As previously described, a Calcein/PI Cell Viability Assay Kit was used to detect cell membrane permeability to determine the degree of cellular focalization. Fluorescence intensity was observed under a fluorescence microscope and quantified using ImageJ software (V 1.8.0) [46].

### 4.5. Western Blotting for Protein Expression

BV2 cells or brain tissue within 2 mm of the hematoma were homogenized and lysed with RIPA lysis solution with phosphorylase and protease inhibitor for 30 min on ice and sonicated. After measuring the protein concentration, a buffer was added to the protein sample, boiled, and set aside. Protein samples (20 µg/well) were added to an SDS–PAGE gel for electrophoresis, and then the proteins were transferred to a PVDF membrane by selecting appropriate membrane transfer conditions according to different molecular weights. The membrane was blocked with 5% skimmed milk for 1 h at room temperature, incubated with primary antibody at 4 °C overnight, washed with TBST 3 times, and then incubated with secondary antibody for 1 h at room temperature. Finally, the results were observed via chemiluminescence, and the bands were analyzed using ImageJ software (V 1.8.0).

### 4.6. Enzyme Linked Immunosorbent Assay (ELISA)

As described in previous studies, cell culture medium was collected for centrifugation, or 100 mg of brain tissue homogenate around the hematoma was taken for processing and diluted for centrifugation according to the kit instructions. There were 3 samples in each group, and each sample was tested 3 times [47].

### 4.7. Statistical Analysis

IBM SPSS Statistics 26 and GraphPad Prism 9 software were used for the statistical analysis. All data were expressed as the mean and standard deviation (SD). Comparison of means among multiple groups was performed using one-way or two-way ANOVA with Bonferroni post hoc test. A value of *p* < 0.05 was considered statistically significant.

## Figures and Tables

**Figure 1 ijms-24-14771-f001:**
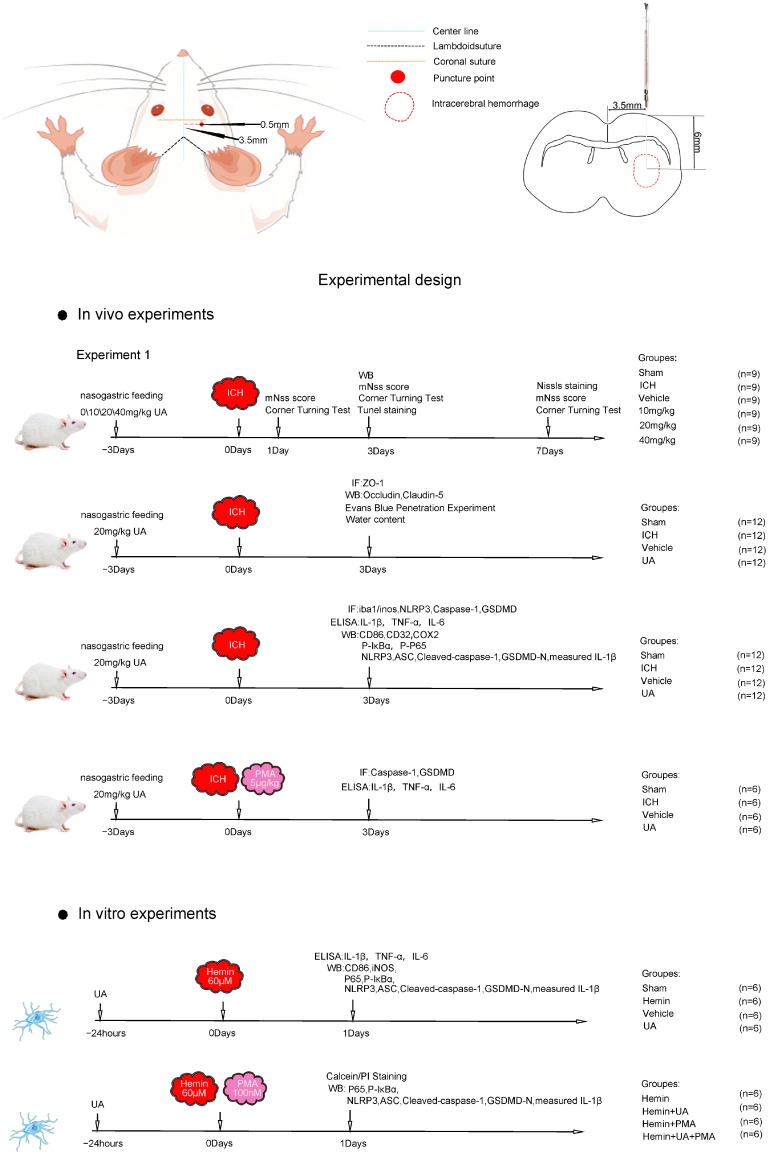
In vivo and in vitro test procedures.

**Figure 2 ijms-24-14771-f002:**
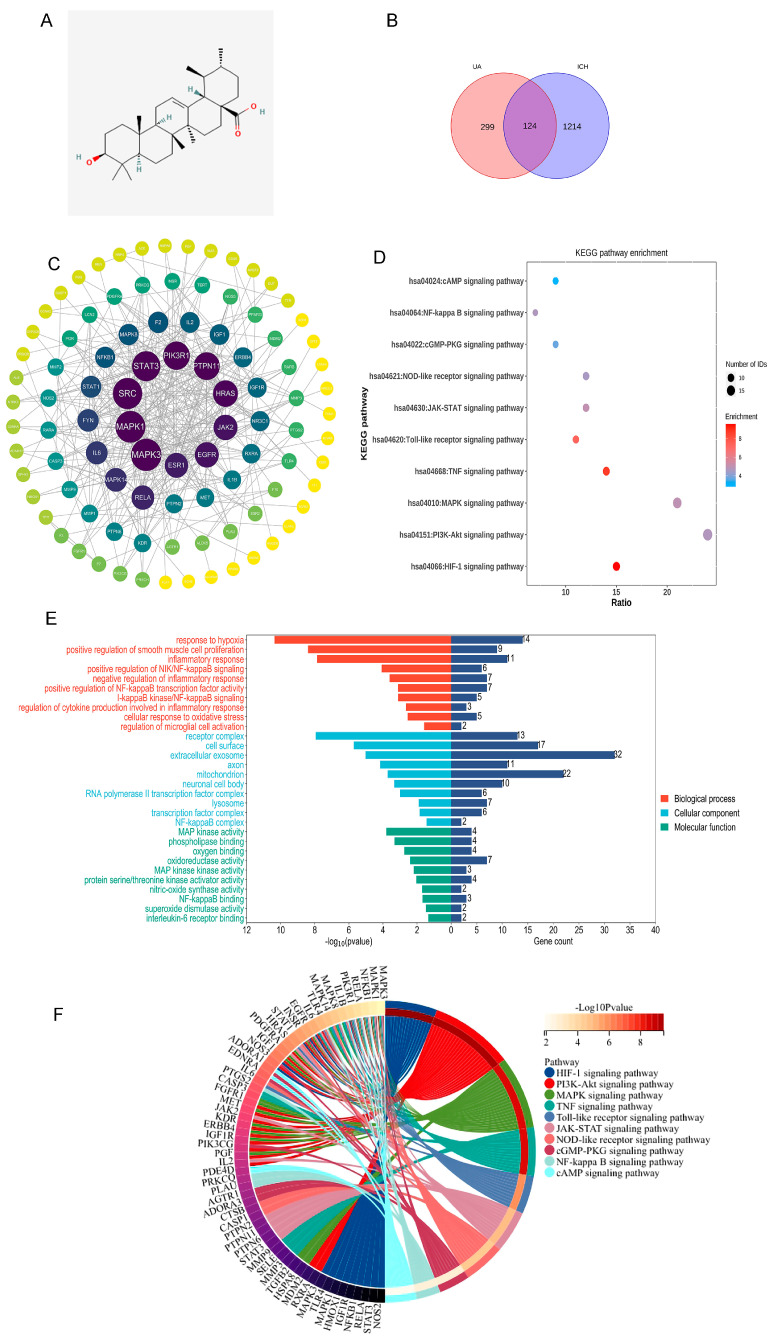
Analyzing the possible pathways of ursolic acid in the treatment of cerebral hemorrhage. (**A**) Ursolic acid molecular structure. (**B**) Venn diagram of potential UA action targets for ICH therapy. (**C**) Protein–protein interaction (PPI) network of potential UA action targets. The circles represent the target protein. The larger the diameter of the circle, the larger the degree value. Straight lines indicate the interaction between target proteins. (**D**) 10 significantly enriched KEGG pathways. (**E**) 10 significantly enriched terms (*p* < 0.05) in biological process, cellular component, and molecular function of GO analysis were selected. The *X*-axis represents the gene count and enrichment score. (**F**) Detailed relationships between differentially expressed genes and major pathways annotated by KEGG are shown by the Circos graph.

**Figure 3 ijms-24-14771-f003:**
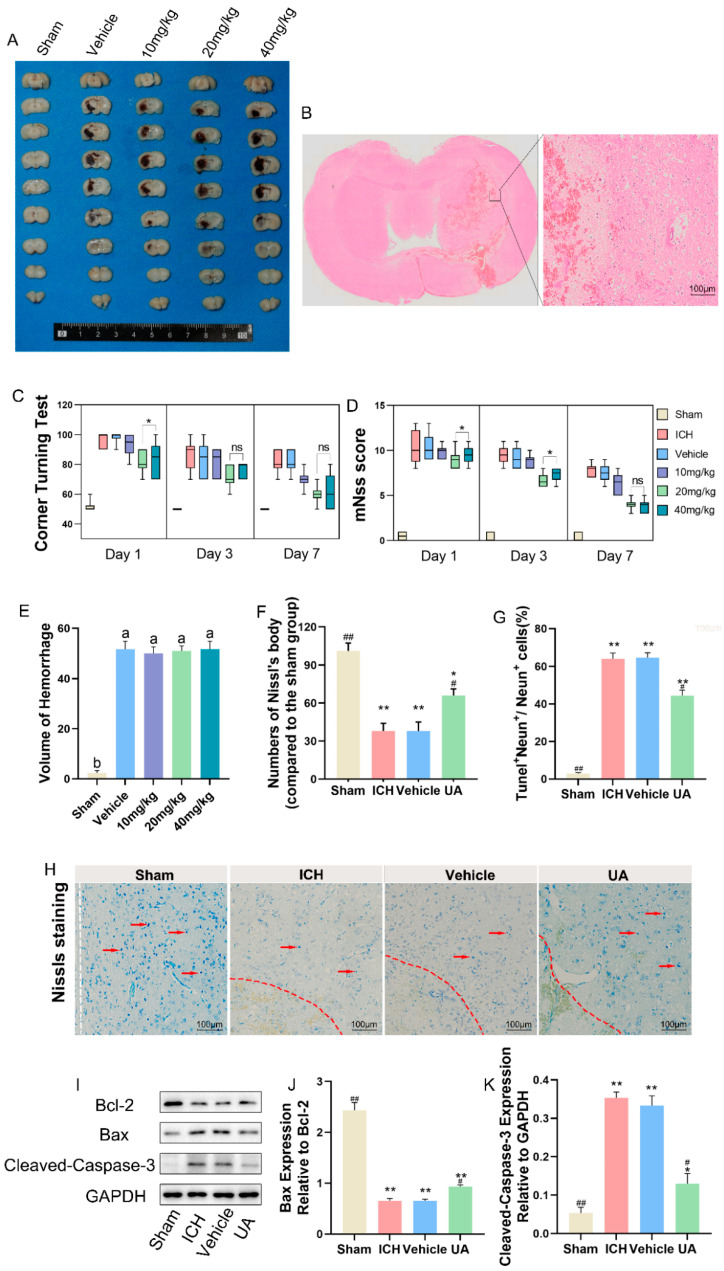
UA exerted neuroprotective effects against intracerebral hemorrhage-induced neurological deficits. (**A**,**E**) Rats were fed with different doses of UA for 3 days and then executed, and brain tissue sections were obtained; ImageJ software (V 1.8.0)was used to calculate the size of hematoma volume in each group, and feeding different doses of UA did not result in differences in hematoma volume. (**B**) HE staining showed cellular edema and necrosis around the hematoma. (**C**,**D**) The Corner Turning Test and the mNSS score were used to assess the improvement of neurological deficits in rats with cerebral hemorrhage after ursolic acid treatment. (**H**,**F**) Nissls staining showed that UA treatment significantly reduced neuronal loss after cerebral hemorrhage. (**G**) Tunel and Neun co-labeling was used to probe the extent of neuronal apoptosis in brain tissue surrounding the hematoma. (**I**–**K**) Bax/Bcl-2, Cleaved-caspase3 expression in different groups. Data are expressed as mean ± SD, scale bar = 100 μm. * *p* < 0.05 vs. Sham group; ** *p* < 0.01 vs. Sham group; ^#^
*p* < 0.05 vs. ICH group; ^##^
*p* < 0.01 vs. ICH group. The red dashed line represents the edge of the hematoma, and the white dashed line represents the needle track.

**Figure 4 ijms-24-14771-f004:**
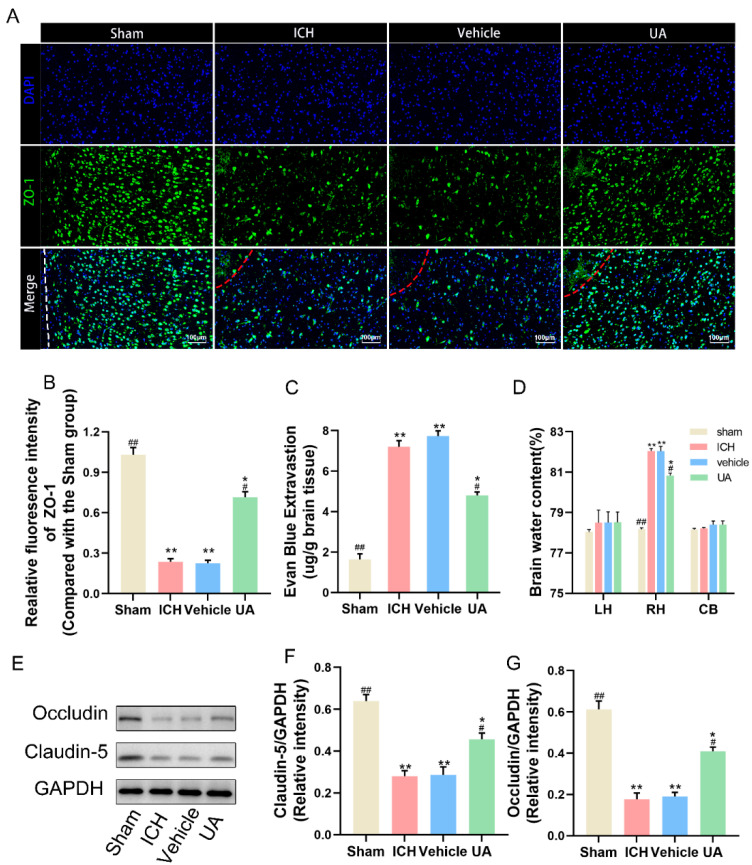
UA treatment attenuates blood-brain barrier disruption by cerebral hemorrhage. (**A**) Immunofluorescence analysis of ZO-1 expression. (**B**) Quantitative analysis of ZO-1 fluorescence intensity. (**C**) Results of Evans blue staining experiments on brain tissues around hematoma of different groups. (**D**) Measurement of brain tissue water content. (**E**–**G**) Representative Western blot expressions of occludin and claudin-5. Data are expressed as mean ± SD, scale bar = 100 μm. * *p* < 0.05 vs. Sham group; ** *p* < 0.01 vs. Sham group; ^#^
*p* < 0.05 vs. ICH group; ^##^
*p* < 0.01 vs. ICH group. The red dashed line represents the edge of the hematoma, and the white dashed line represents the needle track.

**Figure 5 ijms-24-14771-f005:**
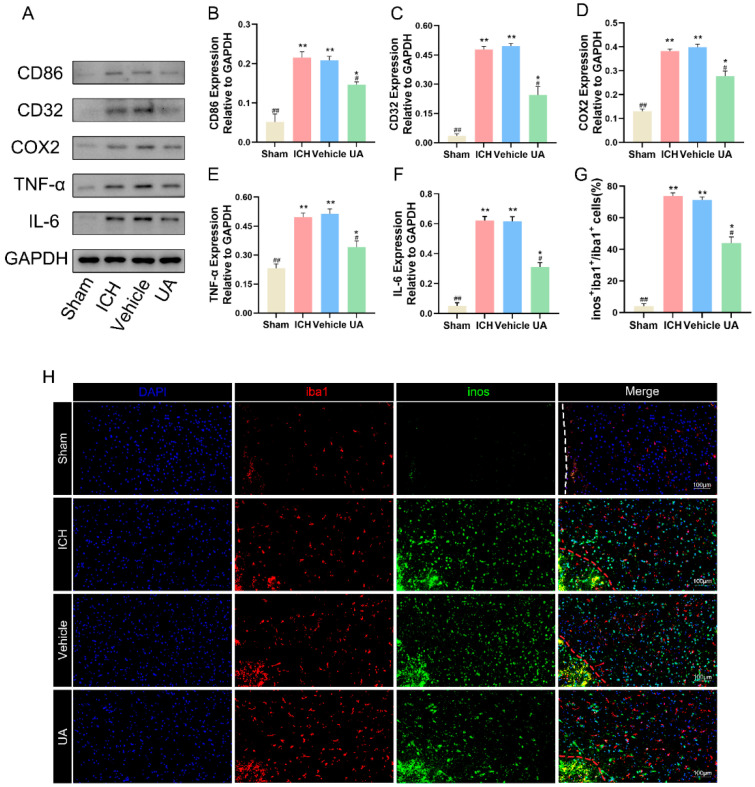
Ursolic acid inhibits the inflammatory response and decreases microglial M1 polarization. (**A**) The protein expressions of CD86, CD32, COX2, TNF-α, and IL-6 in brain tissue surrounding the hematoma were detected by Western blot. Quantitative analysis of the levels of CD32 (**B**), CD86 (**C**), COX2 (**D**), TNF-α (**E**), and IL-6 (**F**) in experimental groups. (**H**) Double immunofluorescent staining was examined around the hematoma of all groups. Iba1 is the microglial cell marker, and inos is the marker for M1-polarized microglia. (**G**) The ratio of inos^+^iba1^+^/iba1^+^ cells. Data are expressed as mean ± SD, scale bar = 100 μm. * *p* < 0.05 vs. Sham group; ** *p* < 0.01 vs. Sham group; ^#^
*p* < 0.05 vs. ICH group; ^##^
*p* < 0.01 vs. ICH group. The red dashed line represents the edge of the hematoma, and the white dashed line represents the needle track.

**Figure 6 ijms-24-14771-f006:**
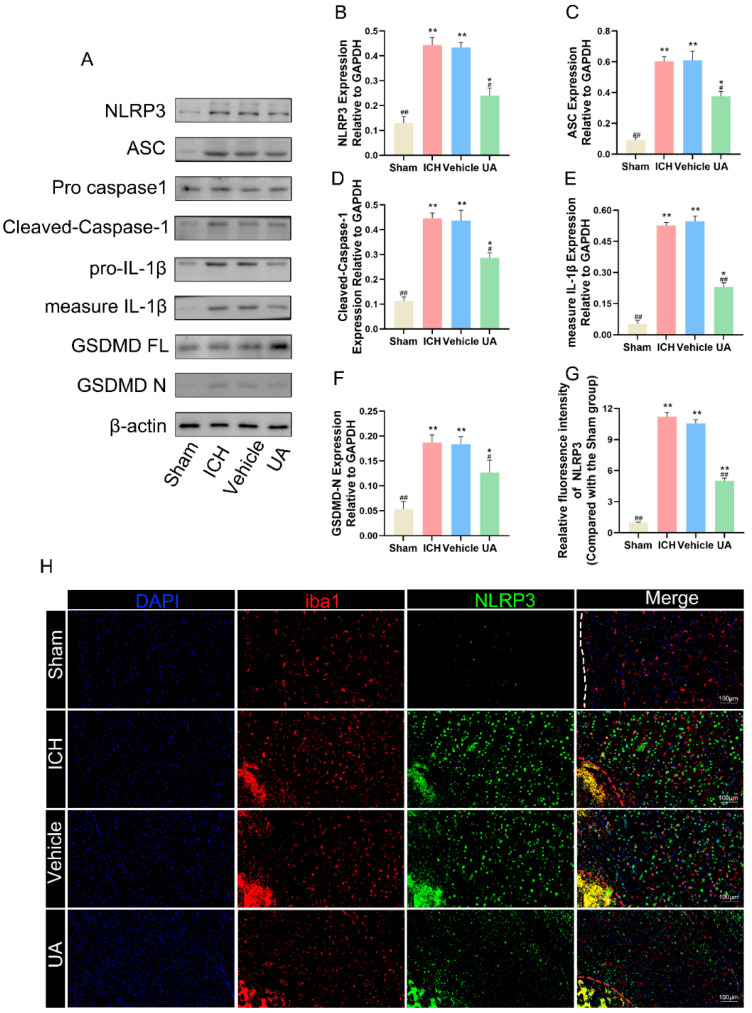
Ursolic acid treatment reduces cerebral hemorrhage-induced microglial cell pyroptosis in vivo. (**A**) The protein expressions of ASC, NLRP3, measure IL-1β, cleaved-caspase-1, and GSDMD-N in brain tissue surrounding the hematoma were detected by Western blot. Quantitative analysis of the levels of ASC (**B**), NLRP3 (**C**), measure IL-1β (**D**), cleaved-caspase-1 (**E**), and GSDMD-N (**F**) in experimental groups. (**H**) Representative immunofluorescent staining of Iba1 with NLRP3 in brain tissue surrounding the hematoma. (**G**) Quantitative analysis of the NLRP3 intensity. Data are expressed as mean ± SD, scale bar = 100 μm. * *p* < 0.05 vs. Sham group; ** *p* < 0.01 vs. Sham group; ^#^
*p* < 0.05 vs. ICH group; ^##^
*p* < 0.01 vs. ICH group. The red dashed line represents the edge of the hematoma, and the white dashed line represents the needle track.

**Figure 7 ijms-24-14771-f007:**
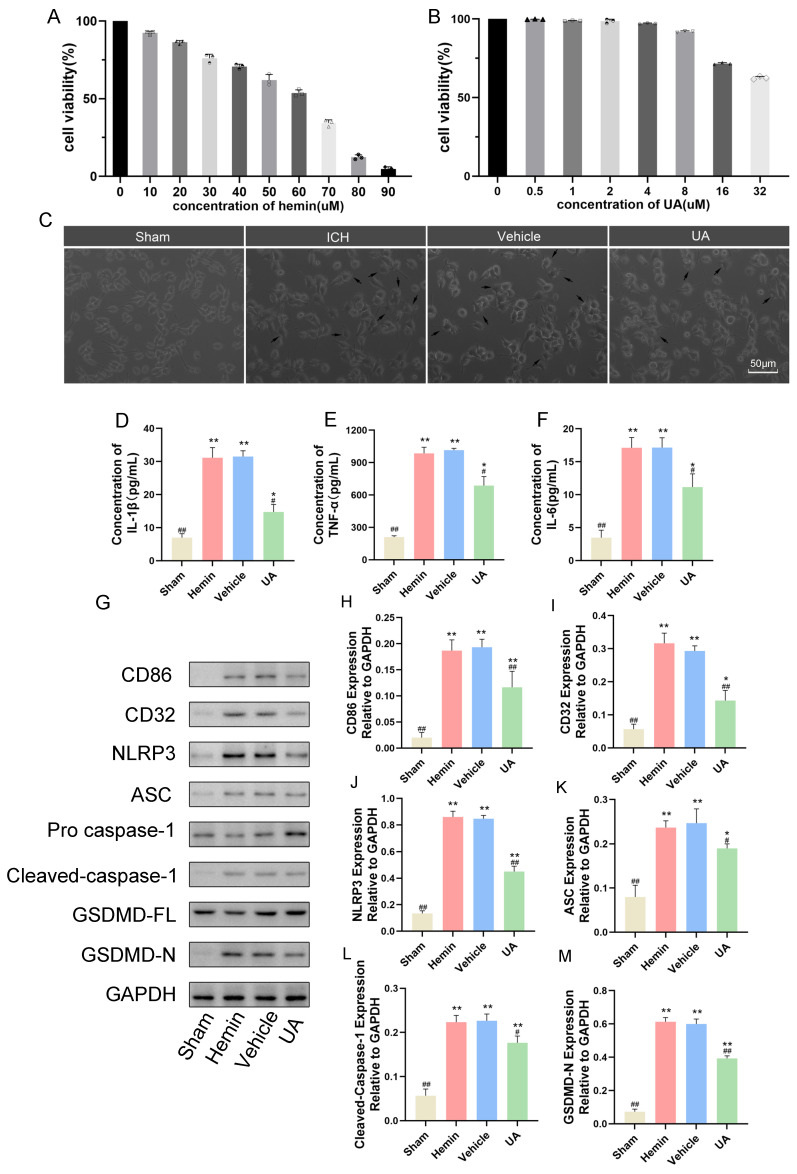
Ursolic acid inhibits the microglial cell inflammatory response and pyroptosis stimulated by hemin in vitro. Cells were treated with (**A**) hemin (0–90 μM) or (**B**) UA (0–32 μM) for 24 h, and then cell viability was measured. The results are expressed as the percentage of values in the control group. (**C**) Growth of microglia under different treatment conditions, with resting microglia having longer or rounded synapses and M1-polarized microglia showing an amoeboid morphology. Black arrows indicate M1-polarized microglia. Quantitative analysis of the levels of IL-1β (**D**), TNF-α (**E**), and IL-6 (**F**) by ELISA kits. (**G**) The protein expressions of CD86, CD32, NLRP3, ASC, cleaved-caspase-1, and GSDMD-N in microglia were detected by Western blot. Quantitative analysis of the levels of CD86 (**H**), CD32 (**I**), NLRP3 (**J**), ASC (**K**), cleaved-caspase-1 (**L**), and GSDMD-N (**M**) in experimental groups. Data are expressed as mean ± SD, scale bar = 50 μm. * *p* < 0.05 vs. Sham group; ** *p* < 0.01 vs. Sham group; ^#^
*p* < 0.05 vs. Hemin group; ^##^
*p* < 0.01 vs. Hemin group.

**Figure 8 ijms-24-14771-f008:**
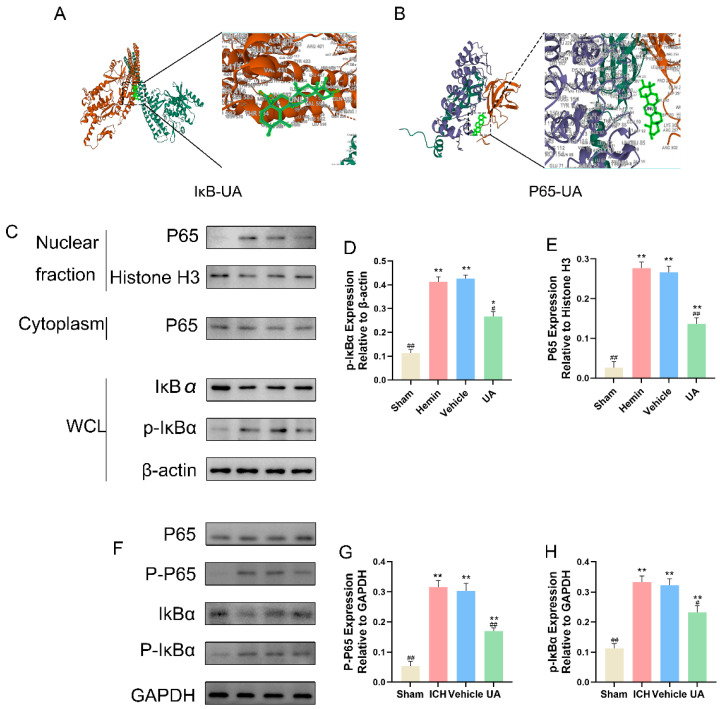
The 3D visualization of molecular docking of UA and NF-κB pathway: IκB-UA (**A**); P65-UA (**B**). (**C**) Western blot analysis of the protein level of NF-κB p65 and IκBα in cell lysates of nucleus and cytoplasm, respectively. The protein expressions of p-IκBα (**D**) and P65 (**E**) in microglia were detected by Western blot. (**F**) The protein expressions of p-P65 and p-IκBα in Brain tissue surrounding the hematoma were detected by Western blot. Quantitative analysis of the levels of p-P65 (**G**) and p-IκBα (**H**) in experimental groups. Data are expressed as mean ± SD. * *p* < 0.05 vs. Sham group; ** *p* < 0.01 vs. Sham group; ^#^
*p* < 0.05 vs. ICH group; ^##^
*p* < 0.01 vs. ICH group.

**Figure 9 ijms-24-14771-f009:**
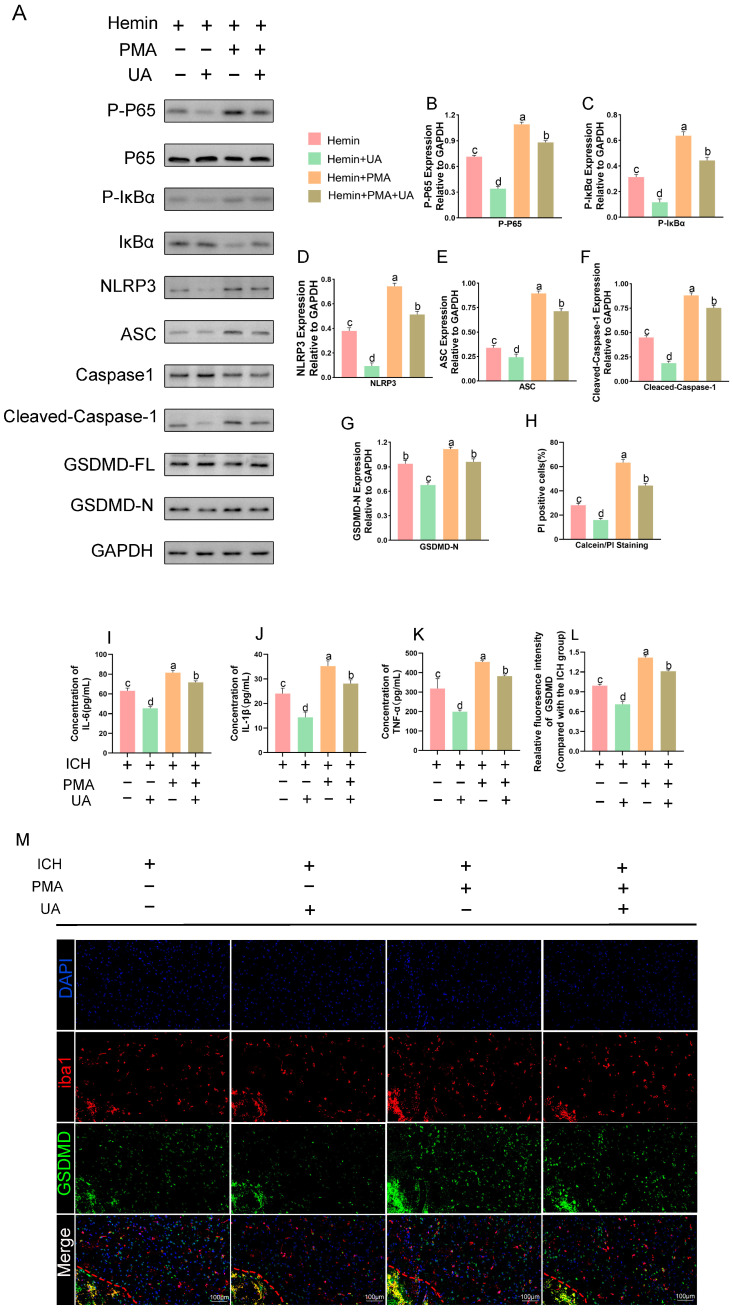
PMA abrogates the role of UA in attenuating microglial pyroptosis. UA attenuates the activation of the NLRP3 inflammasome and the NF-κB pathway in hemin-treated BV2 cells (**A**–**G**). Detection of the degree of BV2 cell pyroptosis using PI staining (**H**). Quantitative analysis of the levels of IL-6 (**I**), IL-1β (**J**), and TNF-α (**K**), and by ELISA kits. Immunofluorescence analysis of GSDMD expression. (**M**) Quantitative analysis of GSDMD fluorescence intensity. (**L**). Data are expressed as mean ± SD, scale bar = 100 μm. Significant differences were analyzed using the “abcd” letter marking method of identification: first, the group means were arranged in descending order, then the largest mean was marked with an a, and that mean was compared to the remaining means. Where the difference is not significant, it is marked with the letter a, until a significant difference with the mean marked with the letter b, and marked with the letter c and d. The difference is not significant in the group marked with the same letter. The difference is significant in the group marked with a different letter. Lowercase letters denote *p* < 0.05, uppercase letters denote *p* < 0.01. The red dashed line represents the edge of the hematoma, and the white dashed line represents the needle track.

**Figure 10 ijms-24-14771-f010:**
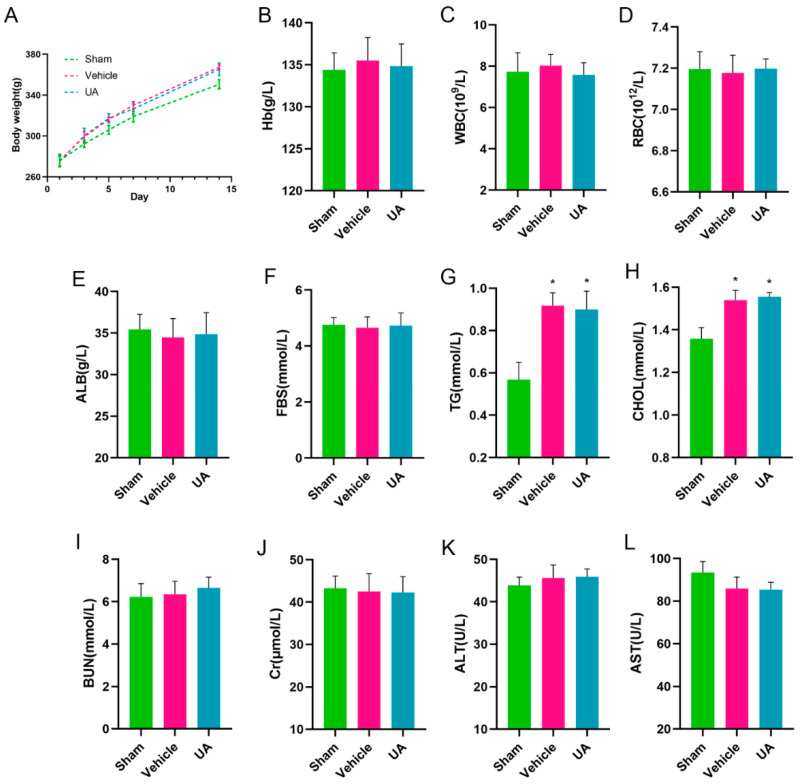
In vivo biosafety assessment of ursolic acid. (**A**) The body-weight curve in 14 days of the rats intravenously injected with UA (20 mg/kg) (*n* = 6). Hematology surveys of the rats show all parameters in a normal range, and there are no significant differences between the groups (**B**–**D**). Indicators of blood nutritional status (**E**–**H**). The blood biochemical test shows the normal function of the kidneys (BUN and Cr) (**I**,**J**) and livers (ALT and AST) (**K**,**L**). The significance of differences was determined using the *t*-test. * *p* < 0.05 vs. Sham group.

## Data Availability

The original contributions presented in the study are included in the article/Appendix A. Further inquiries can be directed to the corresponding author.

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
