# Peer review of "Ursolic Acid Alleviates Neuroinflammation after Intracerebral Hemorrhage by Mediating Microglial Pyroptosis via the NF-κB/NLRP3/GSDMD Pathway"

_ijms, 2023, doi:10.3390/ijms241914771_

Round 1

Reviewer 1 Report

This is a study of Ursolic acid, a natural product present in a variety of plants, to determine effects on inflammation pathways in rat forebrain following intrastriatal hemorrhage. The authors also provide complementary data on the effects of hemin, a protoporphyrin derivative of hemoglobin that is endogenously produced during hemolysis and degradation of “old” red blood cells, on mouse BV2 microglial cells in culture.

The authors first carried out a comprehensive molecular pathways analysis to identify ursolic acid pathways or brain hemorrhage pathways, then combined the two to find significant pathways or Gene Ontology terms involved in both processes. Later, after amassing evidence that the NF-kB pathway was involved in hemorrhage inflammation reduced by ursolic acid, they carried out a molecular docking study to show where ursolic acid could bind to protein pockets of IkB or P65, two major proteins involved in the NF-kB-mediated responses.

To produce intrastriatal hemorrhage, the authors stereotactically injected collagenase (or vehicle controls) into rats that then received ursolic acid (or vehicle), or ursolic acid followed by phorbol-12-myristate-13-acetate (PMA), a NF-kappaBeta agonist. They then carried out survival analyses, motor behavioral tasks and molecular biological studies (mainly Western blots of proteins involved in initiating, maintaining, or executing inflammation). Some Nissl stained and immunohistochemical brain sections are provided. Some complementary studies were carried out in BV2 cells.

The authors concluded that Ursolic acid improves survival, motor function recovery and molecular markers of inflammation in both rats and BV2 cells (where examined), and that this improvement appears to involve modulation of the NF-kB/NLRP3 (a well-studied inflammasome)/GSMD (particularly its cleavage by activated Caspase-1) pathways. Ursolic acid was well tolerated, and preliminary biosafety/toxicology studies showed no detriments. Ursolic acid appears to have substantial therapeutic potential for treatment of brain hemorrhage, particularly reduction of neuronal apoptosis mediated by activated “M1-mediated” inflammation.

This is potentially a very meaningful study, both because of its comprehensive approach and therapeutic implications.

I have the following concerns:

1.    The authors induced intrastriatal hemorrhage by disruption of brain extracellular matrix following collagenase injection. There do not appear to be any references cited as the suitability of this animal model. The authors are likely aware that most intrastriatal (putaminal) hemorrhages in humans are believed to occur from direct rupture of small intracerebral arterioles likely damaged by long-standing hypertension. As a former neurologist I am concerned about the possibility of inflammatory reactions to injected collagenase, not to blood per se. The BV2 experiments tend to support blood (or its products) as the initiators of inflammation, but the authors need to discuss at the minimum why they chose this collagenase model (as opposed to direct blood injection), what limitations their choice places on their conclusions and provide references to use of this model.

2.    Figure 1. Very helpful as presented.

3.    Figure 2. Very helpful as presented, although limited by too small fonts. Could these be enlarged for clarity?

4.    Figure 3.A. Too small to be meaningful. B. Not helpful and can be eliminated or made Supplementary figure. C., D. Need to show significant motor behavior scores to ICH doses alone, and that 20 mg/kg is not different than 40 mg/kg. E.F.G. Fonts too small. H. could be made Supplementary figure. I. Totally not helpful, can be Supplementary figure. J.K.L. Fonts too small, esp. K. and L. Near normalization of cleaved Caspase-3 is an important finding.

5.    Figure 4. A. too light as presented, needs increase in green channel. This should be OK if all images are subjected to the same increase. This could also be a Supplementary figure, since quantitative analysis is provided by (B). B. fonts too small. Important finding. C.D. F.G.  Fonts too small.

6.    Figure 6. A.-G. fonts too small. H. needs enhancement of green intensity for images (could also be Supplementary figure), and would benefit from quantitative analysis like in Figure 4.

7.    7. Figure 7. A.-G. fonts too small. H. Would benefit from quantitative analysis, enhancement of green and red channels. Image of sections could be moved to Supplementary data.

8.    8. Figure 8. A. There is a difference between what is stated in the text (lines 219-226) and what is shown if A. 50% reduction in BV2 cell viability occurs at 40 mM Ursolic acid, not 60 mM as in the text. B. Fine as presented. What is image? C? All pesented bar graphs suffer from too small fonts.

9.    Figure 8. A. B. Fine as presented, although could be a little larger. C. – H. fonts too small.

10. Figure 9. Image I is useless to show PI (+) cells. Bar graph quantitation (H.) is helpful, but all bar graphs suffer from too small fonts. Image (N) needs color enhancement and could be Supplemental figure and benefits from quantitative br graph (M).

11. Figure 10. This entire Figure represents a very preliminary biosafety assessment, which would be very inadequate for use of Ursolic acid in humans, at least in Western countries. Figure 10 should be a Supplemental figure with the results described in the Results text. (or even could be described in Discussion).

12. Figure 11. Very helpful. Please enlarge fonts.

Reviewer 2 Report

The author came up with interesting findings of ursolic acid as an anti-inflammatory after intracerebral hemorrhage.

For the author, I have a few minor  concerns

1. Figures quality should be improved

2. Why I the IF images of NLRP3, iNOS, and ZO-1 looks almost the same

3. In fig 1E-G, why there is no difference between vehicle and different UA doses in 1E whereas, there is in 1F and 1G

4. In Fig 9, the IF image almost seems like background staining. 

Round 2

Reviewer 1 Report

This revision to a manuscript I previously reviewed is much improved compared to the original. I have only one very minor suggestion: I cannot find where the abbreviation "CCK-8" (Cell Counting Kit-8) was ever designated. I confess I may have missed this, but if I am correct, then this abbreviation needs to be added at the appropriate place.

Otherwise, the authors have responded well to all my suggestions about the originalmanuscript. Specifically, as indicated in their response letter, the needed font size increases (and font bolding) have been added to many figures. The immunofluorescent images are also much brighter and thus contributory. The authors discuss their choice of a collagenase model for ICH, compared to stereotactic blood injection. I still wish they would refer to other's use of this model, or if it is being introduced by them, so state it.

As I discussed earlier, the authors provide significant basic mechanistic information about ursolic acid as a potential therapy to improve the neurological outcome in intracerebral hemorrhage. This is the major contribution of this work.